# Physician Perspectives on Malnutrition Screening, Diagnosis, and Management: A Qualitative Analysis

**DOI:** 10.3390/nu16142215

**Published:** 2024-07-11

**Authors:** Daniel Veldhuijzen van Zanten, Erik Vantomme, Katherine Ford, Leah Cahill, Jennifer Jin, Heather Keller, Roseann Nasser, Laura Lagendyk, Tina Strickland, Brenda MacDonald, Sonya Boudreau, Leah Gramlich

**Affiliations:** 1Department of Medicine, Division of Gastroenterology, University of Alberta, Edmonton, AB T6G 2R3, Canada; 2Department of Medicine, University of Saskatchewan, Regina, SK S4P 2H8, Canada; 3Department of Kinesiology & Health Sciences, University of Waterloo, Waterloo, ON N2L 3G1, Canada; 4Department of Medicine, Dalhousie University, Halifax, NS B3J 1V7, Canada; 5Saskatchewan Health Authority, Regina, SK S4P 1C4, Canada; 6QuotesWork, Diamond Valley, AB T0L 2A0, Canada; 7Nova Scotia Health Authority, Halifax, NS B3H 2E1, Canada

**Keywords:** malnutrition, nutrition risk screening, qualitative analysis, malnutrition care and barriers

## Abstract

Malnutrition is an important clinical entity that is frequently underdiagnosed and undertreated, in part due to a lack of education and different perceptions by healthcare providers on its value in medical practice. Given this void, the purpose of this qualitative study was to explore physicians’ clinical perspectives on malnutrition care, including its prevalence in their practice, and potential barriers that might preclude the delivery of malnutrition care. Using a directed content qualitative analysis approach, a total of 22 general and subspecialist physicians across three Canadian provinces were interviewed using a series of standardized questions developed by a multidisciplinary research team. Responses were transcribed and then analyzed using NVivo Version 14 software. While physicians recognized the importance of malnutrition screening and treatment, they did not view themselves as the primary drivers and often deferred this responsibility to dietitians. Lack of standard malnutrition screening, education amongst allied healthcare providers, time, personnel, and referral processes to have patients assessed and managed for malnutrition were also identified as contributing factors. For physicians, malnutrition education, standard malnutrition screening during patient encounters, and access to the necessary tools to manage malnutrition using a more centralized approach and standard referral process were viewed as strategies with the potential to improve the ability of the physician to identify and manage disease-related malnutrition and its negative consequences.

## 1. Introduction

Malnutrition is defined as “both the deficiency and excess (or imbalance) of energy, protein, and other nutrients” resulting in impaired function [1]. Diagnostic clues can be obtained from physical exam (i.e., weight reduction, loss of muscle mass and subcutaneous fat, localized or generalized fluid accumulation, decreased hand-grip strength), biochemical parameters (i.e., prealbumin, albumin, CRP, micronutrients), or screening tools such as the Subjective Global Assessment (SGA) [2]. Malnutrition is often underdiagnosed due to a variety of factors, including a lack of standard nutrition risk screening during patient encounters, a lack of malnutrition education throughout medical training [3,4,5,6], inadequate resources to provide proper nutritional support, and differing perceptions regarding the importance of this clinical condition [7,8]. A systematic review of hospital-acquired malnutrition found that regardless of nutritional status at the time of admission, 10-65% of patients in hospital will experience a decline in nutritional status during their hospital stay [9]. Patients with malnutrition are at risk of increased length of stay, unplanned readmission, and mortality compared with patients without malnutrition concerns [7,10,11,12]. It is therefore vital that the nutritional status of each patient be evaluated at every healthcare encounter. 

To improve clinical outcomes of key nutrition care objectives, the Integrated Nutrition Pathway for Acute Care (INPAC) was developed from current evidence and through consensus with multidisciplinary stakeholders from across Canada [13]. This is similar to efforts that were recently made via a Global Leadership Initiative (GLIM) that was held in January 2016 in an attempt to standardize the clinical practice of malnutrition diagnosis [14]. INPAC is centered on the nutrition care process and addresses malnutrition screening, assessment, intervention, monitoring, and discharge planning. The INPAC was implemented in select hospitals across Canada, including in Nova Scotia, Saskatchewan, and Alberta, as well as other provinces, as part of the More-2-Eat Studies, and now more broadly nationwide with the aim of spreading and scaling improved malnutrition screening, diagnosis, and treatment [15]. The purpose of these studies was to determine if the implementation of INPAC had a positive impact on nutrition care and patient outcomes [15]. As the most responsible provider, physicians are integral to malnutrition care from admission to discharge and can support identifying, managing, and improving the care of patients with malnutrition. As INPAC is being implemented across Canada, it is relevant to determine the perspectives of physicians on their self-perceived role in these malnutrition care domains.

Research that explores physician engagement in the management of nutrition is limited—to the best of our knowledge, only one such study exists. In 2005, Duerksen et al. surveyed 1220 physicians in 18 Canadian hospitals to determine physicians’ perceptions of the detection and management of malnutrition [7]. With a 35% response rate, they found that 29–41% of respondents felt that nutrition assessments were not being performed on a regular basis either at admission, during hospitalization, or at discharge, despite there being a majority consensus that malnutrition care was relevant to patient management [7]. While hospital-associated malnutrition is a common issue with a prevalence as high as 69% [13,15], physicians perceived there to be a wide gap between the optimal detection of malnutrition in hospitalized patients and how it was being identified and managed in hospitals [7]. To further compound this issue, Duerksen et al. identified that the majority of physicians were unfamiliar with the subjective global assessment (SGA), a standardized assessment tool used to identify malnutrition in patients [7]. It was the perception of physicians that patients with malnutrition were not routinely evaluated by dietitians, who identified the need to better integrate nutrition support into patient care including greater access to dietitians to help manage this condition [7]. This was also identified in the Nutrition Care in Canadian Hospitals Study conducted by the CMTF, where only 26.9% of patients with malnutrition were seen by a dietetic professional, contrasted by the fact that patients without these concerns were seen more frequently [1].

With this perceived deficit in malnutrition care by physicians and building on the findings by Duerksen et al., the aim of this qualitative study was to identify the knowledge, perception, and application of malnutrition care by physicians and to identify ways to improve the delivery of malnutrition care [7]. Specifically, we sought to understand how physicians can support nutrition care and detect and treat malnutrition in Canadian hospitals.

## 2. Methods

### Guiding Framework

The Consolidated Framework for the Implementation of Research (CFIR) was first developed by Damschroder et al. in 2009 as a means of understanding and evaluating the factors that influence the successful implementation of interventions and policies [16]. It was updated in 2022 based on user feedback [17]. Using the updated CFIR to develop the interview questions (Appendix A), individual interviews using standardized questions were conducted with physicians from different fields (internal medicine physicians, hospitalists, family physicians, surgeons, and gastroenterologists) based in three Canadian provinces who were involved with some form of nutrition care management within their scope of practice at urban and/or rural sites. Provinces were selected based on jurisdictions in which the two interviewers had familiarity and plans to practice, as well as experience in these provinces with the implementation of best practice in the More-2-Eat studies. This study was approved by Research Ethics Boards at the University of Alberta (PRO00122813) and Dalhousie (1026877). Participants provided written or verbal informed consent prior to participation. The Standards for Reporting Qualitative Research (SRQR) guided the presentation of findings [18].

Purposive and convenience sampling were used to recruit participants along with snowball sampling using targeted emails sent by the multidisciplinary research team and through recommendations from interviewees. Interviews were completed by two trained physician interviewers. Following preliminary analysis and discussions amongst the research group after the initial eleven interviews, the questions were modified slightly in an attempt to improve the quality and breadth of the answers received. These modified questions were posed to the remaining eleven participants and are listed in Appendix A. To increase the generalizability of the findings, interviews were conducted with physicians from a variety of specialties, and with various degrees of inpatient/outpatient practices in provinces across Canada. The research team met to ensure the standardization of interview techniques and to optimize data analysis. Prior to the consensus about interview questions, technique, and follow-up questions, the differences between the two interviewers were harmonized.

Interviews were conducted using Zoom with audio and video recorded and were uploaded to a secure shared folder and transcribed. Initial transcripts were reviewed by the respective interviewer and multidisciplinary research team. Following a preliminary review after several interviews had been conducted, feedback was provided to encourage the interviewer to elicit the depth of response to the interview questions. The verbatim transcripts were imported into NVivo, and identifying information was removed. Demographic information (e.g., specialty and practice location) was collected and included in NVivo to allow for a comparison of themes based on these characteristics.

A directed content analysis approach was used [19]. In directed content analysis, a theoretical framework informs data collection questions and the initial coding structure and definitions [19,20]. As analysis progresses, initial codes are modified and refined, and additional codes are added and regrouped into categories and themes as required to accurately represent the data and answer the study questions [20].

In this study, the initial coding structure was based on the interview questions derived from the CFIR framework. An experienced qualitative analyst first coded each transcript to the initial structure and then added additional codes as required. In consultation with the research team, the initial codes were regrouped into categories of relevance to the study aims. Using a list of common words within a category as search terms, queries were run in NVivo to assess and ensure coding consistency and completeness. Once all of the interviews had been completed and with assistance from the qualitative data analyst, the two interviewers’ datasets were then combined and merged into one NVivo project to contrast and compare categories for similarities and differences and to identify themes. Similar categories were merged, unique categories were retained, and all categories were considered to fit with the themes that were identified across the combined dataset. Exemplar quotes were included for each theme. Themes were examined by demographic characteristics such as those that might be seen when interviewing specialists from the same field of practice. Findings were reviewed and approved by all members of the analysis team, including both interviewers.

## 3. Results

A total of 22 physicians (9 females, 13 males) were interviewed from three Canadian provinces. Participants included six family physicians, one hospitalist, one emergency medicine physician, four gastroenterologists, two geriatricians, two surgeons, two general internal medicine specialists, two medical oncologists, and two rheumatologists. Three major themes that were universally discussed by the physicians were identified as (1) the integration of nutrition care processes into physician workflow, (2) barriers affecting the ability to provide care for patients with malnutrition, and (3) enablers: physicians identified needs and resources. An overview of themes, sub-themes, thematic content, and exemplar quotes is provided in Table 1. 

### 3.1. Integration of Nutrition Care Processes into Physician Workflow

While physicians consistently identified that malnutrition is an important aspect of care, they acknowledged that in the current landscape of inpatient and outpatient settings, physicians’ identification and management of patients with malnutrition remain challenging. A gastroenterologist used an American football analogy to explain their view on the importance of malnutrition identification and the integration of other specialties to provide care: “*The bottom line is…to recognize that there is an issue. Then, we become quarterbacks and refer to people who are more knowledgeable, i.e., the dietitians*”. Physicians viewed themselves as leaders in patient care. Another gastroenterologist explained that given their leadership role, they should advocate for nutrition care to be prioritized, as it is often overlooked: “*The physician’s role is to be a leader on the team, to encourage that this [malnutrition] is just as important as monitoring blood tests, monitoring for infection and other sort of complications in hospital. I think it’s often forgotten about with all of the other tests that are done*”. Similarly, a family physician emphasized the importance of their role and the need to integrate other disciplines into patient care: “*We as physicians have that important role of recognizing any possible factor [for malnutrition], underlying causes, and obviously providing education and being the way that patients will be referred to the specific specialist or health care allies that will be there to help patients.*”

Within the nutrition care process, some physicians felt that they could contribute to screening and assessment and that other specialties (e.g., dietetics and food services) should manage nutrition care interventions, as explained by a surgeon: “*I see my role as being the preliminary point of either identifying it [malnutrition] or screening for it, but a lot of the subsequent care does end up getting managed by other services or by the dietitian*”. Other physicians felt that dietitians played a key role in managing the entire nutrition care process:

*“With the in-patient population we do rely on our dietitians that work on the units…triggering a dietitian consultation on everybody there to identify patients that are at risk. We [physicians] don’t systematically have a screening tool or subject everyone through the Subjective Global Assessment or do anything like that. And in fact, we rely on them [dietitians] often to order the bloodwork”*.[Gastroenterologist].

Overall, physicians viewed dietitians as experts in malnutrition and relied on their knowledge and skillset to operationalize nutrition care. One gastroenterologist suggested that fostering multidisciplinary care and drawing on the expertise of each specialty would be an efficient and effective way to provide care: “*there are trained experts in terms of dietitians that can do this, so perhaps it isn’t really worth our time. Not because we don’t care about it, but because if there’s people that are much better at it and that’s what they do every day…we should work with them*”. Physicians also listed nurses, allied health professionals, and food service workers as others who were able to contribute to the nutrition-related care of patients. All physicians reported referral to a dietitian as standard practice in an inpatient setting.

Despite physicians reporting that they could contribute to malnutrition screening and assessment, most did not report that a systematic approach to malnutrition screening or assessment was integrated into their current practice unless the patient was identified by another healthcare professional as being at risk. Risk factors that triggered malnutrition screening included indicators on physical exam, body mass index, extreme under- or overweight, unintentional weight loss, or disease conditions/comorbidities that are risk factors for malnutrition. An internal medicine physician explained,

*“I think most of the time it [malnutrition] is identified by myself and by my residents as kind of a clinical gestalt [on] general exam. We see patients that are cachectic, obvious signs of the deltoid squaring, the muscle wasting. And just kind of the general appearance. I think we suspect it even more in specific conditions like chronic lung disease, COPD, cancer patients or patients that are near or approaching end-of-life. Those are things that we look at to determine a gross screen for nutritional status”*.

Common biochemical markers that physicians considered in relation to nutrition risk and malnutrition included albumin and prealbumin. Many physicians identified that they had a lack of knowledge and confidence about biochemical markers of malnutrition. Some physicians identified that they relied on other professionals, such as dietitians, to suggest appropriate bloodwork and to then interpret it and identify patients with malnutrition. Although these clinical assessments are components of the SGA, it was apparent that most physicians were not familiar with SGA criteria at all or sufficiently to utilize them in practice. When asked about familiarity with the SGA, many physicians were unaware of the tool. No physician reported the routine use of the SGA in clinical practice, and confidence in its validity wavered. A gastroenterologist said,

*“We don’t have a nutrition focused physician at our site. We’re hoping that changes in the near future. I’ve not seen it [SGA] used. Is it helpful? I know it’s helpful because there’s data on its use and how helpful it is [as a] screening tool, and for follow-up, so it is very helpful. I just haven’t routinely utilized it in my practice. And I’m not aware that any of my colleagues have either”*.

Physicians who were aware of the SGA cited lack of time and resources as reasons for not routinely completing it with their patients, as exemplified by a geriatrician:

*“Certainly, the Subjective Global Assessment is what I would hope that our dietitians are using as a tool for intervening and managing malnourishment in the hospital, or in the outpatient side of things. I will admit that I don’t personally use it because it’s time consuming and you need the specialization from a dietitian to be able to meaningfully impact upon patients the results of the SGA”*.

Physicians recognized that communication between themselves and dietitians can be infrequent or inconsistent, particularly when transitioning care from inpatient to outpatient settings, as described by a geriatrician: “*The majority of the time the interaction is through written consultation notes from the dietitian, but whenever I run into them, I’ll ask them about the patients that I referred to them*”. When asked how physicians follow up with patients with nutritional issues, some physicians reported that they referred inpatients to primary care or community resources after discharge, while others scheduled patients for follow-up in their clinic. A gastroenterologist explained some of the health system challenges impacting malnutrition care by saying,

*“There was really very little support on the outpatient side for nutrition follow-up. There wasn’t a lack of interest, it was just a lack of support by the institution to actually provide enough dietary support. So, what we would usually rely on is giving the patient as much written material as they could manage regarding nutrition at the time of discharge. And then if it was a patient I was particularly worried about, I would try to schedule follow-up with them usually within four to six weeks after discharge”*.

However, primary care physicians reported not being able to follow up on post-discharge nutritional concerns due to a lack of time and resources. They felt incapable of providing adequate nutrition follow-up care even to those patients who were identified as being malnourished, primarily due to a perception of a lack of outpatient resources, including compensation and community dietitians. Nonetheless, they still recognized the importance of nutrition care, as exemplified by a primary care physician: “*For patients who were in the hospital, you need to know that their intake is good once they get home and that they’re maintaining, and everything. Say for example, they have heart failure or something like that, you need to make sure that their fluid intake is appropriate and such*”. These barriers to the optimization of nutrition care were not unique to primary care and the transition-of-care period and highlight an overall deficit in sufficient resources from both an inpatient and outpatient perspective in providing adequate and consistent nutrition care.

### 3.2. Barriers Affecting Ability to Provide Care for Patients with Malnutrition

Barriers to routine nutrition risk screening may lead to patients with malnutrition going undiagnosed and untreated. Commonly cited barriers to optimizing malnutrition care are summarized in Table 2. An example of key barriers to managing the malnutrition care of patients was described by a rheumatologist: “*I think probably a lack of knowledge, a lack of time. You’re being pulled in a bunch of different directions, managing a bunch of different things. And then, to some degree, a lack of resources*.” Physicians consistently identified time management as a major barrier to providing care for patients with malnutrition. They frequently mentioned that nutrition is just one of several priorities competing for their limited time with a patient, and as a result, malnutrition screening and management was frequently deemed a low priority. For example, a medical oncologist said, “*Time. Time. Time and completing other things. There’s so much you have to deal with, within the half-an-hour slot. There is so much to deal with that frankly nutrition is going to fall off the radar or going to be one of the lowest priorities that you have. Unless someone brings about important issues*.” In addition, physicians identified that contact time with inpatients is relatively short, and time was a barrier to implementing the use of a standardized nutrition care process:

*“…how do you implement it [SGA] when you have no time? …because a lot of these assessments take two-to-three-to-five minutes and we’re already maxing out the patients in clinics and so we tack on the amount of time…realistically, dietary needs, sexual needs, there’s so much that needs to be looked at when we book patients and we do a poor job because we just don’t have the time to do that proper assessments for patients”*.-Medical Oncologist

Many of the physicians identified a lack of education and experience and competing patient factors as barriers to identifying, managing, and prioritizing malnutrition. A family medicine physician explained, “*It’s a lack of education for sure…in medicine…we always keep learning something new [about medications], but about nutrition, not so much…I think physicians think it is important, but not as important as the medical conditions. Yes, I think it [nutrition] is in second place*”. Another family medicine physician explained the challenge with prioritizing malnutrition: “*Sometimes we are biased towards recognizing patients with malnutrition…we look at a patient and we look at why they are admitted…hardly ever do we identify nutritional status as a cause of the problems, but more as a consequences of a main admission reason*”. Physicians often attributed insufficient education about nutrition to their medical school and residency training. One gastroenterologist explained, “*In gastroenterology we’re probably more knowledgeable than any medical specialties in terms of nutrition, but are we tooled to manage it? I would say, no, at least most of us aren’t. We learn it in residency and then and then quickly forget it*”. Primary care providers and other specialists explained that malnutrition care was not adequately taught in their residency and fellowship to prepare them for independent practice. In most cases, physicians identified that this lack of education led to nutritional issues being overlooked and under-recognized, as exemplified by a family medicine physician: “*I don’t pay attention unless it’s very obvious that the patient is malnourished. Otherwise it’s just not relevant or not as important because there are other issues that are more important*”. In another instance, a physician identified that a treatment culture that emphasized pharmacotherapy over other interventions was not conducive to effective nutrition care.

Other barriers to malnutrition care that were identified by physicians included access to adequate resources and the complexity of nutrition management. In the inpatient setting, a general and colorectal surgeon explained, *“I’m sure we could improve a lot of patient’s outcomes if we were able to take a couple weeks and work on their nutrition as an inpatient. We just don’t have the resources to allow for that*”. The quality of hospital food itself can pose a barrier to providing care as unappetizing meals lead to reduced intake. If patients do receive nutritional support in the hospital and long-term supplementation upon discharge, the cost of these products can be prohibitive to effective care. Similarly, food insecurity and insufficient social support, including access to a social worker to address patients’ food security concerns, were identified as a barrier. Lastly, in some areas, physicians described access to dietitians as a limiting factor in collaboration that hinders timely and coordinated nutrition care.

*“I think nutrition is essential. I’m not sure how much impact we have in a short stay. It’s usually one-to-three days that people are with us. As an impatient we can refer them to the dietitian…. And as an outpatient we have offered, but we don’t usually get a lot of uptake because it’s cumbersome to book. They’re not booking with an in-hospital nutritionist who’s at the clinic where we’re at, they would have to be booking in an outpatient’s clinic setting”*.Family Medicine Physician.

Barriers identified by physicians such as lack of knowledge, education, time, or competing patient priorities that are instead addressed through guided thoughts on enablers to support malnutrition care.

### 3.3. Enablers: Physicians Identified Needs and Resources

Physicians identified multiple areas of need when asked about resources to assist with the prevention, detection, and treatment of malnutrition; see Table 2. Overall, they felt that a multidisciplinary approach to nutrition care would benefit the patient but that a streamlined approach was necessary. One medical oncologist explained, “*If you…see what a cancer patient has to go through from an assessment point of view, it’s a lot. They get bombarded with questions and I know if they get asked more questions, they just kind of shut down… If you…call them to see how they’re doing at home, that’s outside the clinic, that would probably help…[provide] better care for patients*”. The importance of streamlining care to reduce patient burden went beyond assessment processes to enhancing access for patients, as noted by a family medicine physician: “*Having easy access to a nutritionist, but more-so a social worker…that is just integrated into the clinic, it’s not something…that they have to book another appointment for or arrange*”. To support the integration of a multidisciplinary approach to care, physicians felt that an algorithm that included additional support or a referral process for triaging patients and connecting them with outpatient resources upon discharge would be beneficial.

When asked if they were aware of any existing care pathways or resources that they could integrate into practice, most physicians were unaware of such tools, as suggested by a medical oncologist: “*Integration of some sort of relatively straight forward applicable clinical decision [tool]…into daily practice, if not done so already. Something that nurses screen for regularly…as part of a clinical assessment that can sort of red flag those who are at highest risk*”. The creation of an inpatient and outpatient pathway that would clearly demarcate where patients should be referred for assistance based on established criteria was emphasized by multiple physicians as a method of improving nutrition care. “*There could be some improvement in our communication from the specialist level to primary care the nutritional requirements of the patients, especially in a world where I can only see them once a year*”. (Gastroenterologist). This would include social and socioeconomic support to help address food insecurity if required for patient care. Throughout the discussion, emphasis on the desire for evidence of effectiveness and data to highlight the impact of nutrition care were exemplified, as noted by a geriatrician: “*Give me all the validated tools. Give me all the research that shows why I need to do this so I can really dig in and also impart buy-in*.”

Many physicians wanted additional education for the efficient identification of malnutrition from an interdisciplinary perspective. A family medicine and emergency physician explained, “*It’d be neat to get the entire unit or team on board…the doc, the nurses, pharmacists, dietitians, everyone all on board and have some sort of an education session on malnutrition so that it’s more of a team approach*”. Others suggested including education about ordering and the assessment of lab results. There was also a request for physical resources for patients, such as pamphlets or handouts, that would provide additional patient education about what to monitor at home regarding their nutritional status and what they could try if they began to show signs or symptoms of malnutrition. In-person or online educational sessions for patients were also mentioned by some physicians.

## 4. Discussion

This is the first study to review physicians’ perceptions of malnutrition screening, diagnosis, and management since the implementation of INPAC in Canadian hospitals. The results of this study suggest that primary care physicians and specialists recognize that malnutrition is an issue that is overlooked due to various factors, including a lack of education, knowledge, skill, and training to consistently identify malnutrition, as well as resources.

Furthermore, despite their role as the most responsible provider, physicians did not view themselves as the drivers of malnutrition care, citing time constraints and competing patient priorities, and instead identified dietitians as leaders in this area. Similar to findings from Rigling et al., food insecurity was mentioned as an important factor contributing to malnutrition [21,22,23,24,25]. A lack of community social supports throughout the patient trajectory from home to hospital was also cited as a contributing factor to malnutrition, which is similar to findings in other studies [26,27]. Physicians also recognized that a medical culture that prioritized pharmacotherapy can hinder nutrition-related care. Similar to Dent et al., physicians spoke about the positive effect of viewing “food as medicine” and the importance of collaboration with a multidisciplinary team including dietitians, nursing, social work, and food services [28,29]. Communication, particularly when transitioning care from the inpatient to outpatient setting, was identified as an opportunity for improvement.

Findings from this qualitative study demonstrate key themes and challenges faced by physicians as they attempt to navigate effective and optimal nutrition care. In contrast to Duerksen et al.’s survey that also explored physician perspectives of malnutrition, our qualitative study involved physicians working in both the inpatient and outpatient settings from a variety of disciplines, including primary care and medical/surgical specialties. Physicians consistently acknowledged that while recognizing and treating malnutrition was important for patient care, they did not have adequate training, knowledge, or screening tools to consistently identify and treat malnutrition.

In existing nutrition care pathways [1], physicians are identified as key healthcare professionals who can diagnose malnutrition. However, routine screening for malnutrition is not commonly practiced by physicians, and most physicians are not aware of the SGA, the gold standard diagnostic tool for assessing nutritional status and diagnosing malnutrition. Time constraints and resource limitations were identified as reasons why physicians do not use the SGA in practice. In addition, physicians felt that their undergraduate and postgraduate medical education did not adequately equip them with the knowledge needed to engage with standardized screening tools for malnutrition, nor were physicians aware of the benefits of optimized nutrition care. Nonetheless, physicians were able to recognize weight loss, body mass index, muscle wasting, or physical indicators (general appearance, loose skin, etc.) as a means of identifying signs of malnutrition.

Strategies to enhance the physician’s role in nutrition care included standardized care pathways for both inpatients and outpatients that detail the role of the physician and timely referral to a multidisciplinary team and community partners [1,30,31]. Other approaches included enhancements in education through learning modules, didactic presentations, pamphlets, or handouts, as identified by the participants in this study, including evidence that assessment and treatment processes improve nutritional status. Physicians also cited patient-oriented materials as a means to help empower patients to monitor their nutritional status and seek appropriate care when needed. Other approaches suggested by physicians included multidisciplinary rounds that include dietitian involvement with standardized reporting protocols for reporting the oral intake, nutrition status, and nutrition intervention, to optimize communication. This would also improve the ability of physicians to intervene earlier in the clinical course for patients at a high risk of developing malnutrition. A summary of this is provided in Table 2, using the COM-B model, which provides a behavioral change framework that demonstrates the interaction between Capability, Opportunity, and Motivation to necessitate a behavioral change [32]. Per the model, one must have the capability and opportunity to engage in the behavior, and the strength of motivation to engage in the behavior must be greater than any other competing behavior [33]. This framework was chosen as it provides a relatively simple way for various stakeholders to take action in various areas to improve the delivery of nutrition care.

The strengths of this study include interviewing a geographically diverse set of practicing physicians in several different disciplines from primary care providers to specialists. Limitations possibly include purposive sampling, as those physicians interviewed may have an interest in nutrition care. Future directions could include a follow-up study with the same and/or additional participants to see if the identification and management of malnutrition has improved in the areas identified and highlighted through this study.

## 5. Conclusions

Nutrition care in hospitalized patients, including malnutrition screening, diagnosis, and intervention, is a human right [26,34]. Physicians acknowledged the importance of malnutrition and recognized that through interdisciplinary teams including knowledgeable dietitians, they had a deeper understanding of the importance of nutrition care in acute settings and during transitions of care. Physicians acknowledged that improvements in health systems’ processes such as screening and embedded care pathways would support improved patient care. Physicians are in a position to foster multidisciplinary care and can build on the strengths of the dietitian and standardized processes to nutrition risk screening and assessment to address malnutrition care.

## Figures and Tables

**Table 1 nutrients-16-02215-t001:** Themes that describe physicians’ (*n* = 22) role in malnutrition care.

Theme	Sub-Themes	Thematic Content	Exemplar Quotes
Integration of Nutrition Care processes into physician workflow	Screening	Identifying patients with malnutrition is crucial, and physicians can play a key role	“*The physician is running what’s happening, they should be pushing the rest of the team to maximize nutrition in the patients*”.
			Gastroenterologist
		Routine screening is rarely performed by physicians; they rely on allied healthcare, including dietitians, to identify patients with malnutrition	“*We just sort of say, ‘this patient’s at risk because they have a chronic GI condition, consult a dietitian’, and then we rely on them to identify the patients at risk*”
			Gastroenterologist
		Physicians do not routinely use standardized screening tools	“*I’ve personally never used [the SGA] once in my practice. I’ve not seen [the SGA] used in any of my colleagues’ practices.*”
			Gastroenterologist
	Assessment	Common clinical markers were weight loss, BMI, muscle wasting, or visual indicators	“*We see patients that are cachectic, obvious signs of the deltoid squaring, the muscle wasting*”.
			Internal Medicine Specialist
		Common biochemical markers were albumin and prealbumin.	“*We look at albumin, prealbumin. That’s our kind of easy marker of where you’re at. There’s probably better ones out there”*.
			General & Colorectal Surgeon
	Intervention	Physicians viewed their role as identifying and referring patients with malnutrition, rather than treating	“*I see my role as being the preliminary point of either identifying malnutrition or screening for it*”.
			Surgeon
	Monitoring	Outpatient follow-up for nutrition is rare	“*There was very little support on the outpatient side for nutrition follow-up*”.
			Gastroenterologist
Barriers to providing care for patients with malnutrition	Time	Physicians prioritize other medical problems over malnutrition screening and management	“*Physicians are incentivized to spend as little time as possible with patients so [malnutrition] is something that’ll often get overlooked*”.
			Geriatrician
	Lack of knowledge and education	Physicians feel poorly equipped with knowledge and clinical exposure to handle malnutrition	“*…this subject is not taught in undergraduate medical school…it’s given a cursory nod*”.
			Gastroenterologist
	Lack of sufficient resources	Inadequate number of dietitians to help meet malnutrition goals	“*Lack of resources…and easy access to dietitians…creates some barriers to referral in follow-up.*”
			Primary Care
	Food insecurity	Rising inflation impacts affordability of healthier food options	“*Making sure that they have access to food, that they can afford food. Getting social work involved if they can’t.*”
			Family Medicine & Hospitalist
Enablers: physician-identified needs and resources	Multidisciplinary care	Involvement of multiple healthcare professionals to improve malnutrition care	“*…it’s more of a team approach verses just a siloed approach*”
			Family & Emergency Medicine
	Clinical pathways	The development of standardized malnutrition pathways would improve the delivery of care	“*…some sort of relatively straight forward applicable clinical decision [tool] that could be integrated into daily practice*”
			Internal Medicine
	Evidence of efficacy	Demonstrable proof that malnutrition care leads to improved patient outcomes	“*…an idea of what to track metric-wise then you could try and see if what we actually did matters or actually made any difference*”
			Family & Emergency Medicine
	Continuing medical education	Improved education for physicians about how to screen for and manage malnutrition	“*…we could do a better job in terms of educating family doctors or at least providing direction as to what they should be measuring…*”
			Gastroenterologist
	Financial	Due to financial constraints and food insecurity, less expensive options for patients would assist in providing consistent care.	“*…there’s definitely a gap in terms of the patient’s ability to pay for [oral nutritional supplements and nutritional support] and the expense*”
			Gastroenterologist

**Table 2 nutrients-16-02215-t002:** Barriers and Potential Solutions to Implementing Malnutrition Care Identified by 22 Physicians.

Barrier	Factor of COM-B Model for Behavioral Change	Potential Solution
Limited communication between physicians and dietitians	Opportunity	Establish regular meetings and routine use of multidisciplinary rounds to discuss nutrition related patient care. Making a diagnosis of malnutrition on chart documentation and discharge summaries will improve consistency in care.
Lack of familiarity with validated screening tools including the SGA	Capability	Provide routine education and training at the undergraduate and postgraduate levels along with continued medical education (CME) opportunities.
Time constraints and competing patient priorities	Capability	Incorporate malnutrition screening as an admission standard, or on a yearly basis for outpatients, and ensure food services employees and dietitians communicate daily with physicians; dietitians available in primary care.
Lack of access to affordable nutritional supplements and medical food (i.e., oral nutrition, supplement)	Opportunity	Advocate for policies and drug plans that promote access to nutritional supplements for patients. Emphasize that food is medicine.
Limited social support and food insecurity	Opportunity	Collaborate with social workers and community resources as a standard part of nutrition care.
Limited multicultural menu choices for hospitalized patients	Opportunity	Increase the number of multicultural dishes that patients can choose from; encourage patients to bring food from home.
Absence of clear, standardized referral pathways for nutrition care	Opportunity, Motivation	Develop an algorithmic approach and referral pathway that utilizes local resources to guide physicians in connecting patients with appropriate outpatient resources.

## Data Availability

The original contributions presented in the study are included in the article, further inquiries can be directed to the corresponding authors.

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
