# Peer review of "Physician Perspectives on Malnutrition Screening, Diagnosis, and Management: A Qualitative Analysis"

_nutrients, 2024, doi:10.3390/nu16142215_

Round 1

Reviewer 1 Report

Comments and Suggestions for Authors

Dear Authors,

The topic of your manuscript is significant as malnutrition, especially among older adults is a prevalent condition and even considered one of Geriatric Giants. In many countries screening for malnutrition is compulsory at hospital admission. However, as you mention, physician engagement in the management of nutrition is limited. Unfortunately, in everyday practice, a proper malnutrition screening is not done regularly either at admission, during hospitalization, or at discharge.

Here are my questions or suggestions:

1.       Is one of the keywords, i.e. ”qualitative” correct? Or maybe you meant qualitative analysis?

2.       Please search your paper for some minor grammar mistakes and typos, for instance in line 63 period should be after the bracket. The same situation refers probably to all references;

3.       More information about the diagnosis of malnutrition would be welcome (more than using screening tools), including the GLIM consensus for the senile population;

4.       In my opinion, despite the study is based on a detailed protocol and is very work and time-consuming it should be still considered preliminary, which should be emphasized in the title.

I am looking forward to your answers.

Best regards,

The reviewer.

Author Response

  1. Is one of the keywords, i.e. ”qualitative” correct? Or maybe you meant qualitative analysis?

Thanks. I have made this change in the document. It is highlighted in red.

2. Please search your paper for some minor grammar mistakes and typos, for instance in line 63 period should be after the bracket. The same situation refers probably to all references;

Thanks. Formatting was intended to have the numbers be superscripts which I thought would appear after the period. However, I have gone through the document and changed all the reference numbers so that they appear before periods. This is now in line with the journal guidelines.

3. More information about the diagnosis of malnutrition would be welcome (more than using screening tools), including the GLIM consensus for the senile population;

Thanks. I have attempted a few sentences (highlighted in red) in two sections under the "Background" section that hopefully addresses this. (Lines 50-54, 67-70).

4. In my opinion, despite the study is based on a detailed protocol and is very work and time-consuming it should be still considered preliminary, which should be emphasized in the title.

Thanks. I have added the word "Preliminary" in the title.

Reviewer 2 Report

Comments and Suggestions for Authors

Thanks for submitting your manuscript to Nutrient. You can find my comments below.

Abstract

    ●No need to bold the headings

    ●If you won't mention DRM again, delete the abbreviation

Keywords

    ●According to the study purpose, the keywords should include malnutrition care and barriers

Main content

    ●The citation format shoukd folliw the journal instruction, using [xx] before the period '.' 

    ●Well written

    ●Line 137, what information was identified and removed? 

    ●Lines 197 and 218 as well as the others afterward, revise formating. Font and font size should follow the journal instruction. 

    ●E.g., Line 391, the reference number should be used instead of the year according to the journal. Please check the whole manuscript. 

    ●References: #1 Full name goes first and then blanket the abbreviation

    ●Should follow the journal instruction. Please check all the reference list. 

    ●The authors should follow the template provided by the journal for all sections. After the references, there are some sections, such as Appendix, to be completed. 

Author Response

Thanks for submitting your manuscript to Nutrient. You can find my comments below.

Abstract

  • No need to bold the headings
  • If you won't mention DRM again, delete the abbreviation

Thanks. I have unbolded the headings and removed the DRM abbreviation.

Keywords

  • According to the study purpose, the keywords should include malnutrition care and barriers

Thanks. I have added this in red text to the manuscript.

Main content

  • The citation format shoukd folliw the journal instruction, using [xx] before the period '.'  Thanks. I have fixed all of these in-text citations throughout to follow this format.
  • Well written

  • Line 137, what information was identified and removed? The physician's practice both in terms of discipline and geographic location were removed so as to now bias the qualitative analysis.

  • Lines 197 and 218 as well as the others afterward, revise formating.

In reviewing the Instructions for Authors, I  am uncertain as to how to alter this as these are quotes from interview participants and was unable to locate any information on what format this should follow.  

  • Font and font size should follow the journal instruction. 

In reviewing the Instructions for Authors, I am unable to find details on font type and size. I have therefore not altered these.

  • E.g., Line 391, the reference number should be used instead of the year according to the journal. Please check the whole manuscript. 

I have fixed this and other instances where the in-text citation was incorrect (i.e. line 77, 86, 96, 391, 396, 403).

  • References: #1 Full name goes first and then blanket the abbreviation

I have attempted to update this using the guidelines provided in the Information for Authors but reference #1 does not have an author.

  • Should follow the journal instruction. Please check all the reference list. 

I have attempted to update this using the guidelines provided in the Information for Authors. References have been updated with changes noted in red font.

  • The authors should follow the template provided by the journal for all sections. After the references, there are some sections, such as Appendix, to be completed. 

The Appendix was not included in the revisions so I am unsure what you are referring to. I did download a copy of the Supplementary Tables and reformatted the titles slightly based on the Instructions to Authors.